# Early Intervention and Lifelong Treatment with GLP1 Receptor Agonist Liraglutide in a Wolfram Syndrome Rat Model with an Emphasis on Visual Neurodegeneration, Sensorineural Hearing Loss and Diabetic Phenotype

**DOI:** 10.3390/cells10113193

**Published:** 2021-11-16

**Authors:** Toomas Jagomäe, Kadri Seppa, Riin Reimets, Marko Pastak, Mihkel Plaas, Miriam A. Hickey, Kaia Grete Kukker, Lieve Moons, Lies De Groef, Eero Vasar, Allen Kaasik, Anton Terasmaa, Mario Plaas

**Affiliations:** 1Laboratory Animal Centre, Institute of Biomedicine and Translational Medicine, University of Tartu, 14B Ravila Street, 50411 Tartu, Estonia; kadri.seppa@ut.ee (K.S.); riin.reimets@ut.ee (R.R.); kaiagretekukker@gmail.com (K.G.K.); anton.terasmaa@kbfi.ee (A.T.); 2Department of Physiology, Institute of Biomedicine and Translational Medicine, University of Tartu, 19 Ravila Street, 50411 Tartu, Estonia; eero.vasar@ut.ee; 3Eye Clinic of Tartu University Hospital, L. Puusepa 8 Street, 50406 Tartu, Estonia; marko.pastak@kliinikum.ee; 4Ear Clinic of Tartu University Hospital, L. Puusepa 1a Street, 50406 Tartu, Estonia; mihkel.plaas@kliinikum.ee; 5Department of Pharmacology, Institute of Biomedicine and Translational Medicine, University of Tartu, 19 Ravila Street, 50411 Tartu, Estonia; miriam@ut.ee (M.A.H.); allen.kaasik@ut.ee (A.K.); 6Research Group Neural Circuit Development and Regeneration, Department of Biology, Belgium & Leuven Brain Institute, University of Leuven, Naamsestraat 61, Box 2464, 3000 Leuven, Belgium; lieve.moons@kuleuven.be (L.M.); lies.degroef@kuleuven.be (L.D.G.)

**Keywords:** wolfram syndrome, Wfs1, GLP1 receptor agonist, liraglutide, rat model, neurodegeneration, optic nerve atrophy, hearing loss, visual acuity, diabetes

## Abstract

Wolfram syndrome (WS), also known as a DIDMOAD (diabetes insipidus, early-onset diabetes mellitus, optic nerve atrophy and deafness) is a rare autosomal disorder caused by mutations in the Wolframin1 (*WFS1*) gene. Previous studies have revealed that glucagon-like peptide-1 receptor agonist (GLP1 RA) are effective in delaying and restoring blood glucose control in WS animal models and patients. The GLP1 RA liraglutide has also been shown to have neuroprotective properties in aged WS rats. WS is an early-onset, chronic condition. Therefore, early diagnosis and lifelong pharmacological treatment is the best solution to control disease progression. Hence, the aim of this study was to evaluate the efficacy of the long-term liraglutide treatment on the progression of WS symptoms. For this purpose, 2-month-old WS rats were treated with liraglutide up to the age of 18 months and changes in diabetes markers, visual acuity, and hearing sensitivity were monitored over the course of the treatment period. We found that treatment with liraglutide delayed the onset of diabetes and protected against vision loss in a rat model of WS. Therefore, early diagnosis and prophylactic treatment with the liraglutide may also prove to be a promising treatment option for WS patients by increasing the quality of life.

## 1. Introduction

Wolfram syndrome (WS) is a rare autosomal recessive disorder caused by mutations in the Wolframin 1 (WFS1) gene. WS is characterized by juvenile-onset insulin-dependent diabetes mellitus, diabetes insipidus, deafness, optic atrophy, neurological complications and endocrine abnormalities [1,2,3,4,5,6,7,8]. Therefore, WS is established as a spectrum disorder where the severity and emergence of the features depend on the genetic background and location of the mutations which lead to the dysfunction of the WFS1 protein [4,5,6,7,8]. In WS patients, the diabetes can be effectively managed, leaving the loss of vision and neurodegeneration as the main symptoms impairing the quality of life [8]. Thus, the development and evaluation of neuroprotective strategies is of utmost importance from the patient’s perspective.

The WFS1 gene encodes an 890 amino acid long transmembrane protein (WFS1 or Wolframin) that is localized on the endoplasmic reticulum (ER) membrane and in the secretory granules [8,9]. WFS1 protein is highly expressed in the heart, lungs, placenta, pancreas, inner ear, specific regions of the brain [10,11,12], and in the retina [13]. Its exact function remains elusive, but elevated endoplasmic reticulum (ER) stress, dysregulation of intracellular calcium metabolism [14,15] and dysfunction of mitochondria [16,17] are cellular events connected with functional WFS1 deficiency.

A Wfs1-deficient rat model of WS (Wfs1 KO) was recently constructed and validated at the University of Tartu [18]. Wfs1 KO rats develop a more prominent diabetic phenotype than any previously generated mouse model of WS and show neurodegeneration of the brainstem and optic nerve. As such, this rat model of WS recapitulates the clinical findings observed in human patients and therefore provides an excellent model to evaluate pharmacological treatment strategies [18,19,20,21].

Glucagon-like peptide 1 receptor agonists (GLP1 RA) have been approved as a treatment option for type 2 diabetes [22]. The beneficial effects of GLP1 RA on glycemic control are established by modulation of the ER stress, the main contributor to WS [23]. Previous studies have revealed GLP1 RA to be effective in the restoration of glucose control in genetic mouse [24,25] and rat [21] models of WS. Moreover, treatment with the GLP1 RA liraglutide provided neuroprotective and anti-inflammatory effects in aged Wfs1 KO rats by halting the development of optic atrophy and reducing neuro-inflammation [19,20].

WS is a lifelong condition and, therefore, any pharmacological treatment of WS patients will also be lifelong. The GLP1 receptor agonist treatment is currently the only known treatment strategy that was shown to be effective in several animal models of WS and in human patients [21,26]. However, the long-lasting effect of such GLP1 RA treatment in the model of WS has never been evaluated. Hence, the aim of this study was to assess the efficacy of long-term, prophylactic GLP1 RA treatment on the main symptoms of WS using the WS rat model. For this purpose, changes in diabetes markers, visual acuity and hearing sensitivity were monitored in vivo during the course of a 16-month-treatment period with liraglutide.

## 2. Materials and Methods

### 2.1. Animals

All experimental protocols were approved by the Estonian Project Authorization Committee for Animal Experiments (No. 138, 18 March 2019), and all experiments were performed in accordance with the European Communities Directive of September 2010 (2010/63/EU).

Generation and phenotype of a Wfs1 mutant (Wfs1-ex5-KO232, Wfs1 KO) rat has been described previously [18]. Breeding and genotyping were performed at the Laboratory Animal Centre at the University of Tartu. For this study, male homozygous Wfs1-deficient (Wfs1 KO) and littermate control rats (WT) were used. Male rats were chosen as the phenotype and progression of WS has been well established and documented [18]. The animals were housed in cages in groups of 2–4 animals per cage under a 12 h light/dark cycle (lights on at 7 a.m.). To minimize retinal phototoxicity, animals were maintained at dim light conditions (60–80 lux). Rats had unlimited access to food and water except during testing. Sniff universal mouse and rat maintenance diet (Sniff #V1534) and reverse osmosis-purified water were used. Experiments were performed between 9 a.m. and 5 p.m.

### 2.2. Long-Term (Prolonged) Treatment with GLP1 Receptor Agonist Liraglutide

Animals were 2 months old at the start of the treatment and randomly divided into four experimental groups with at least 20 animals in each group: wild type animals with saline treatment (WT Sal), *n* = 23; wild type animals with liraglutide treatment (WT LIRA), *n* = 20; Wfs1 KO animals with saline treatment (KO Sal), *n* = 27; Wfs1 KO animals with liraglutide treatment (KO LIRA), *n* = 24. The liraglutide-treated animals received 0.4 mg/kg liraglutide (Novo Nordisk, Denmark), and the control groups received a 0.9% NaCl solution (saline). Injections of 1 mL/kg volume were made subcutaneously once a day, between 8 and 11 a.m., for 16 months. 

Rats were weighed once a week, and their base blood sugar level was measured once a month from the tail vein using a handheld glucometer (Accu-Check Go, Roche, Mannheim, Germany). In order to avoid hyperglycemia-induced symptoms, supportive insulin treatment (100 IU/mL, Levemir, Novo Nordisk, Bagsværd, Denmark) was initiated in hyperglycemic Wfs1 KO rats. Animals with a basal blood glucose level of 10 mmol/L received 1 IU/kg insulin twice per day by subcutaneous injections. Every subsequent 5 mmol/L increase of blood glucose level, the given insulin dose was increased by 1 IU/kg (i.e., animals with a blood glucose level of 20 mmol/L received 6 IU/kg insulin twice per day) [27,28].

As expected in such a prolonged treatment study, some animals were lost during the experiment or had to be sacrificed for ethical reasons. The number of animals lost in each group was: 7 out of 23 in WT SAL; 7 out of 27 in KO SAL; 8 out of 20 in WT LIRA and 4 out of 24 in KO LIRA. In the vast majority of cases, the cause of death was cancer, which is a common trait of Sprague−Dawley rats [29]. 

### 2.3. Glucose Tolerance Tests (IPGTT)

Animals were deprived of food for 3 h before and during the experiment; water was available throughout the experiment. Glucose (Sigma-Aldrich, now Merck KGaA, Darmstadt, Germany) was dissolved in 0.9% saline solution (20% *w*/*v*) and administered intraperitoneally at a dose of 2 g/kg of body weight. Blood glucose levels were measured at the indicated time points from the tail vein using a hand-held glucometer (Accu-Check Go, Roche, Mannheim, Germany), immediately before and 30 min after glucose administration. Samples were used for insulin (CrystalChem cat# 90060, Elk Grove Village, IL, USA), c-peptide (CrystalChem cat# 90055, Elk Grove Village, IL, USA) and glucagon (CrystalChem cat# 81519, Elk Grove Village, IL, USA) measurements using ELISA tests. For serum separation, blood was allowed to clot, centrifuged for 15 min at 2000× *g*, and stored at −80 °C until further analysis.

IPGTT was performed at the beginning of experiment before the drug treatment (age 2 months) and at the ages of 7.5 and 11.5 months. To minimize the possible acute effects of drug treatment, the IPGTT experiment was performed 24 h after administration of liraglutide or vehicle.

### 2.4. Two-Hour-Postprandial Blood Sugar Test

In order to measure postprandial blood glucose levels rats were deprived of food for 3 h and blood glucose levels were measured from tail vein using glucometer (Accu-Check Go, Roche, Mannheim, Germany). Subsequently blood glucose levels were measured 2 h after the animals had access to food.

### 2.5. In Vivo Magnetic Resonance Imaging

In vivo magnetic resonance imaging (MRI) imaging was performed at the age of 17 months as described previously by our group [18]. For the optic nerve + chiasm + tract, segmentation began where the optic nerves emerged through the optic canal foramina and continued until the point where the optic tract was no longer discernable from surrounding parenchyma. 

### 2.6. Hearing Evaluation

Hearing was evaluated by measuring otoacoustic emissions (OAEs). OAEs reflect the function of outer hair cells [30] and are widely used as a hearing loss screening method in every day clinical practice [31]. A commercial clinical instrument (Sentiero Diagnostics, PATH medical GmbH, Germering, Germany) with frequency modulated distortion product otoacoustic emission (DPOAEs) tympanometry upgrade (FMDPOAETM) was used to measure DPOAEs. DPOAE FULL test protocol was used, including enabled automated background noise calibration, automated threshold detection algorithm and automatic scissor paradigm options. The F2 frequencies where 1, 1.5, 2, 3, 4, 5, 6, 8 kHz. Stimulus level L2: 25 to 65 dB SPL, step 5dB (automated threshold detection algorithm). L2/L1 relation was adjusted automatically by the device software. Cochlear hearing levels were estimated from measured DPOAEs at distinct hearing frequencies using a software protocol preset by the manufacturer [32]. Rats were tested at the age of 2, 6.5, 11.5 and 17 months and all tested ears were inspected visually to rule out potential causes for conductive hearing loss (cerumen, inflammation, fluids, etc.) as DPOAEs cannot distinguish between conductive and sensorineural hearing loss. During the procedure, rats were anesthetized using isoflurane (for induction 3%, and for anesthesia maintenance 1.5% isoflurane was used, flow speed 2 l/min medical oxygen). To avoid anesthesia effects on cochlear outer hair cell and cochlear amplifier function, only the right ear was tested first and animals were woken up [33]. After one hour wash out period, the procedure was repeated to measure the left ear. The average of right and left ear readings was used for analysis. It should be emphasized that DPOAEs only reflect outer hair cell functionality and therefore are not present at a hearing loss higher 50 dB HL. Therefore, hearing loss greater than 50dB in any given frequency is estimated by the software [32].

### 2.7. Visual Acuity Analysis

To measure visual acuity, a virtual optomotor task (OptoMotry, Cerebral Mechanics Inc., Alberta, Canada) was used and the test was performed as described previously [20]. Briefly, animals were placed in the center of a virtual rotating cylinder displaying vertical bars. The width of the bars was made progressively smaller and the tracking behavior of the animals was recorded. The minimal width of the bars that induced tracking behavior is a surrogate measure of the maximal spatial frequency resolution of the rat. Clockwise rotation was detected by the left eye and anticlockwise rotation by the right eye. Visual acuity data are presented as the mean of clockwise and anticlockwise testing. Visual acuity was evaluated at the ages of 8.5, 12.5 and 17 months.

### 2.8. Cataract Scoring

Lens changes, monitored by a portable slit lamp following dilation with 1% tropicamide, were evaluated by a trained ophthalmologist blinded to the experimental protocol, as described previously [34]. Severity of cataract was scored as follows: 0, clear; 1, posterior subcapsular (PSC) haze; 2, spokes, suture enhancement and powdery PSC deposits; 3, prominent PSC punctate opacities with spokes and suture enhancement; 4, PSC opacity blocks appearance of retinal vessels; 5, start of cortical involvement; and 6, total lens opacity visible with naked eye. The mean of both eyes’ cataract scores was calculated and presented. Cataract scores were evaluated at the age of 13 and 17 months.

### 2.9. Intraocular Pressure Scoring

Intraocular pressure (IOP) was measured using an Incare TONOLAB (Vantaa, Finland) device that is specially designed for small rodent (rat/mouse) IOP measurements. During the procedure, rats where anesthetized using isoflurane (induction: 3%; maintenance: 1.5%; flow speed 2 l/min medical oxygen). The IOP for both eyes was measured following the manufacturer’s test protocol. The mean IOP values for both eyes was calculated and presented. IOP was measured at the age of 17 months.

### 2.10. Tissue Collection and Preparation

Rats were deeply anesthetized using a mixture of ketamine (100 mg/kg i.p.) and dexmedetomidine (0.5 mg/kg i.p.) until loss of reflexes. Blood samples (about 10 mL each) was taken via cardiac puncture and thereafter animals were transcardially perfused with 0.1 M phosphate buffer (PB: 100 mM Na_2_HPO_4_, 100 mM NaH_2_PO_4_, pH 7.4) followed by 4% paraformaldehyde in PB (100 mM Na_2_HPO_4_, 100 mM NaH_2_PO_4_, pH 7.2; 300 mL/5 min). Tissues of interest were dissected and immersion fixed in 4% PFA/PBS (PBS, 150 mM NaCl, 36 mM Na_2_HPO_4_, 14 mM NaH_2_PO_4_, pH 7.4) overnight at 4 °C. The PFA solution was decanted and then replaced with fresh PBS supplemented with 0.01% sodium azide, this solution was changed once daily for a period of three days. Thereafter, tissues were cryoprotected in 30% sucrose/PBS solution for 4–5 days at 4 °C with gentle agitation. Tissues were then frozen by immersing in isobutane cooled to −40 °C and kept at −80 °C until further use. 

Optic nerves were carefully removed and the right optic nerve was processed and stored as described above. The left optic nerve was immersion fixed in 2% glutaraldehyde/160 mM sucrose/90 mM sodium cacodylate in water and kept over 24 h at 4 °C for electron microscopy analysis.

### 2.11. Retinal Wholemount Immunohistochemistry

After perfusion, one eye from each animal was enucleated and immersed in 1% PFA/PBS for 12 h and subsequently washed at least 3 times with PBS over 3 days. Retinas were then dissected and kept in PBS until further use. Retinal ganglion cells were visualized and counted via immunostaining for the POU-domain transcription factor Brn3a that is specifically expressed in retinal ganglion cells [35], as previously described in detail by Geerarets et al. [36]. The immunostaining effectively labeled retinal ganglion cells in every treatment group and genotype (Appendix A). Analyses were performed on eight frames, randomly selected in the mid-peripheral and peripheral retina, with a total area of 122,601 um^2^ per retinal wholemount (5–6 animals per condition), and the number of Brn3a positive cells per mm^2^ was calculated.

### 2.12. Electron Microscopy

After the immersion fixation, the optic nerves from each animal (*n* = 7–9) were incubated at 4 °C until further washing with 90 mM Na-cacodylate buffer with subsequent processing as described previously [20]. Electron micrographs for analysis were collected using an Orius SC200B CCD camera (Gatan Inc., Warrendale, PA, USA) mounted on a of Tecnai G^2^ Spirit TWIN/BioTWIN transmission electron microscope (FEI Europe B.V, Eindhoven, The Netherlands). Representative figures for illustrations were collected using a side mounted Veleta CCD camera (Olympus).

### 2.13. Morphometric Evaluation of Optic Nerve Fibers

Morphometric analysis of optic nerve fibers was performed on 35.1756 µm (width) × 23.4504 µm (height) images (pixel size: 0.0458 × 0.0458 µm^2^) using ImageJ software version 1.53c [37]. Cross-sections of optic nerves were divided into 10–11 non-overlapping random regions to minimize the variation in fiber size distributions. Five fields of view were randomly selected for analysis and on average 1400 nerve fibers were counted from individual rats (7–9 animals for each group). Only fibers whose contour was completely within field of view were counted.

Axon area measurements were performed on scale reduced images (scale: x = 0.1918, y = 0.1918). Average axon area per field of view was calculated as the sum of the individual axon areas per field of view. Average myelin areas were measured as the area of grayscale from converted to black, on 8-bit images (threshold 1–255). An area-based g-ratio (axon area divided by axon area plus myelin area) was calculated. The methods for counting degenerating fiber abnormalities were described previously [20].

### 2.14. Data Analysis

GraphPad Prism version 9 software (GraphPad Software Inc., San Diego, CA, USA) was used for statistical analysis, and *p* < 0.05 was considered statistically significant. The data are presented as mean ± SEM and were compared using repeated measures ANOVA followed by Bonferroni’s multiple comparisons test or the data were compared using one- or two-way ANOVA followed by Tukey’s multiple comparisons test.

## 3. Results

### 3.1. A 16-Month Treatment with Liraglutide Delays the Onset of Diabetes in Wfs1 KO Animals

Previously we have shown that a 5-month-preventive treatment with liraglutide, starting before the onset of WS symptoms, protected against the development of glucose intolerance in Wfs1 KO rats [21]. Whether liraglutide prevents or only delays diabetes in Wfs1 KO rats is still unknown, thus the aim of this study was to investigate if long-term, preventive treatment, was able to stop the development of diabetes.

First, body weight changes were monitored over time. Liraglutide treatment induced a slight reduction of body weight during the first week of administration in WT and Wfs1 KO animals (Figure 1a), in agreement with our previous work using GLP1 receptor agonists in this rat model of WS [19,20,21].

Next, to monitor glucose tolerance over time, intraperitoneal glucose tolerance tests (IPGTT) were performed at the age of 2, 7.5 and 11.5 months. At the age of 2 months, there were no differences between the experimental groups (Figure 1b,c). At the age of 7.5 months, saline-treated Wfs1 KO rats had developed glucose intolerance, which was confirmed by an area under the curve analysis (*p* < 0.0001) (Figure 1b,d). Liraglutide treatment prevented the development of glucose intolerance in Wfs1 KO animals (*p* < 0.0001) (Figure 1b,d). At the age of 11.5 months, saline treated Wfs1 KO animals remained glucose intolerant (*p* < 0.0001) (Figure 1b,e) whereas liraglutide treatment was effective against the development of glucose intolerance (*p* < 0.001) (Figure 1b).

Moreover, at the age of 7.5 months, liraglutide-treated Wfs1 KO animals divided into two groups (Figure 1b): one group, who remained diabetes-free as their glucose tolerance was at the same level with WT saline treated animals (63.2%, 12 out of 19 animals) and second group, who developed glucose intolerance similarly to saline treated KO animals (36.8%, 7 out of 19 animals). At the age of 11.5 months, 62.5% (10 out of 16) of the liraglutide-treated Wfs1 KO animals remained diabetes-free.

To prevent complications due to hyperglycemia, supportive treatment with insulin was initiated in 12-month-old Wfs1 KO animals (Figure 1a,f,g). At 17 months of age, only 55% of the liraglutide-treated Wfs1 KO animals required supportive insulin treatment, whereas 90% of the saline-treated Wfs1 KO animals needed supportive insulin (Figure 1g). At the age of 16 months, most of the Wfs1 KO animals had developed hyperglycemia (Figure 1f), therefore it was not ethical to perform IPGTT. Hence, a 2 h postprandial blood sugar test was performed, showing that the blood sugar test results did not differ between treatment groups of Wfs1 KO animals.

To assess the responsiveness of pancreatic islet cells to a glucose challenge, serum c-peptide, insulin and glucagon levels were measured before and 30 min after administration of glucose in the IPGTT at the age of 2, 7.5 and 11.5 months. At the age of 2 months, there were no differences between experimental groups (Figure 2a–c). At the age of 7.5 months, Wfs1 KO rats showed the first signs of diabetes by reduced c-peptide (Figure 2d) and insulin (Figure 2e), and increased glucagon secretion (*p* < 0.01) (Figure 2f). Liraglutide treatment prevented these changes (*p*-value). At the age of 11.5 months, Wfs1 KO rats were not able to secrete c-peptide (Figure 2g) nor insulin (Figure 2h) and their glucagon levels were increased (*p* < 0.001) (Figure 2i). Liraglutide-treated Wfs1 KO rats remained capable of secreting c-peptide (*p* < 0.05) (Figure 2g). Thus, we can conclude that liraglutide treatment delays but does not prevent the development of diabetes in Wfs1 KO rats.

### 3.2. Liraglutide Treatment Did Not Prevent Progressive Sensorineural Hearing Loss of Wfs1 KO Rats

Sensorineural hearing loss in WS patients is highly prevalent therefore we sought to evaluate the hearing sensitivity in WS rats during the course of liraglutide treatment. Cochlear hearing levels were estimated by DPOAEs at the age of 2, 6.5, 11 and 17 months. At the age of 2 months, hearing sensitivity was indistinguishable between experimental groups (Figure 3a). At the age of 6.5 months, the hearing sensitivity of the Wfs1 KO rats started to decline from 2 kHz (*p* < 0.0001) to 5 kHz (*p* < 0.05) (Figure 3b), compared to saline-treated WT rats. The estimated hearing loss further progressed by the age of 17 months both in terms of estimated hearing levels and frequencies affected for the KO rats compared to WT rats (Figure 3d). We found no effect of liraglutide treatment on estimated hearing levels. Although we observed statistically significant differences at 6.5 months of age in liraglutide-treated Wfs1 KO rats (from 2 kHz range) the obtained results serve minimal value (changes less than 10 dB in few individual frequencies). In summary, we found that Wfs1 KO rats lose hearing sensitivity around 6.5 months of age and liraglutide treatment was not able to ameliorate the decline from the analyzed frequencies (Figure 3c).

### 3.3. Sixteen-Month Treatment with Liraglutide Protects against Vision Loss in Wfs1 KO Animals

The complications in the ophthalmic system in WS patients are described as a decrease in visual acuity, constriction of the peripheral visual field and loss in color vision led by optic disc and nerve atrophy [38]. We have revealed that Wfs1 KO rats display clear degenerative processes in the ophthalmic system and late intervention with liraglutide, starting in 9-month-old animals can maintain already reduced visual acuity by delaying the progression of optic nerve atrophy and a potentially regenerating effect on optic nerve fibers [18,20]. Here, our aim was to investigate if life-long treatment with liraglutide can protect Wfs1 KO animals from vision loss.

The optomotor responses were measured at the age of 8.5–9; 12.5 and 17 months (Figure 4). Visual acuity, as measured by the optokinetic response, was lower in 8.5-month-old Wfs1 KO rats as compared to WT rats of the same age (*p* < 0.01) (Figure 4a), and treatment with liraglutide prevented this decline (*p* < 0.001) (Figure 4a). At the age of 12.5 and 17 months, visual acuity of Wfs1 KO animals had steadily declined with age (*p* < 0.0001) (Figure 4a) and this decline was completely rescued by liraglutide treatment (*p* < 0.0001) (Figure 4a). Intraocular pressure did not differ between the experimental groups at any time point measured (Figure 4b). Cataract severity evaluated in 13- and 17-month-old animals, was increased at the age of 13 months in Wfs1 KO rats compared to WT littermates (*p* < 0.001) (Figure 4c), and liraglutide treatment was able to protect Wfs1 KO animals from developing cataracts (Figure 4c). Additionally, a positive correlation was observed between insulin dose (IU/kg/day) and cataract score in saline-treated Wfs1 KO rats (R^2^ = 0.396, *p* < 0.01, Appendix A) and in liraglutide-treated Wfs1 KO rats (R^2^ = 0.271, *p* < 0.05, Appendix A). Thus, the protection from cataract (Figure 4c) results from the therapeutic effect of liraglutide and not from the supportive treatment of insulin. In conclusion, despite insulin supplementation, saline-treated Wfs1 KO animals progressively showed visual decline from 8.5 months of age, yet a 16-month treatment with liraglutide was effective in protecting Wfs1 KO animals against visual loss.

### 3.4. Sixteen-Month Treatment with Liraglutide Protected against the Development of Optic Nerve Atrophy in Wfs1 KO Animals

To unravel the structural changes underlying the vision loss in WS rats, and the protective effect of liraglutide hereon, we next assessed optic nerve integrity and retinal ganglion cell survival. Transmission electron microscopy (TEM) on optic nerves revealed that the number of optic nerve axons per field of view (FOV) was increased (*p* < 0.05) (Figure 5e) while axon size was decreased (*p* < 0.05) (Figure 5f) in liraglutide-treated Wfs1 KO animals, compared to age-matched saline-treated Wfs1 KO rats. The optic nerve axonal area per FOV remained relatively stable in saline- and liraglutide-treated WT rats (Figure 5g). The axon area per FOV was decreased (*p* < 0.001), myelin area per FOV was not changed (Figure 5h), and hence the g-ratio (axon area divided by axon area plus myelin area) was lower (*p* < 0.001) (Figure 5i), in saline-treated Wfs1 KO rats compared to WT littermates. These changes were rescued by the treatment with liraglutide.

Furthermore, the electron microscopy analysis of degenerating optic nerve fibers in Wfs1 KO animals demonstrated several pathological features (Figure 5b). First, abnormalities in the myelin sheath were observed, including both infoldings and outfoldings that typically occurred in large caliber fibers and which resulted in axonal compression. Axonal decompression due to abnormally expanded periaxonal spaces was also seen. Second, degenerating axons appeared hypermyelinated and a complete loss of axons resulted in the remaining myelin bodies (Figure 5b). Third, severe cases of myelin lamellae decompaction were observable independently from axon caliber. Finally, additional pathological signs of axon degeneration included severe vacuolization and axoplasm filled with an amorphous, granular and dark material, in axons with various calibers. These results are consistent with those previously published by our group [19,20]. Semiquantitative assessment of these observable pathological alterations in the myelin and axonal structure further confirmed that axonal pathology was significantly increased in saline-treated Wfs1 KO animals compared to WT animals (*p* < 0.01) (Figure 5j). Treatment with liraglutide reduced the number of degenerating optic nerve fibers compared to saline-treated Wfs1 KO animals (*p* < 0.05) (Figure 5j).

Previously, we have seen a decrease in the volume of the optic nerve in 15-month-old Wfs1 KO animals [18]. Here, in vivo MRI revealed that optic nerve volume was also decreased in saline-treated Wfs1 KO animals at the age of 17 months (*p* < 0.001) (Figure 5k) and the 16-month treatment with liraglutide protected the optic nerve from atrophy.

Additionally, retinal ganglion cell density was measured by counting Brn3a-positive cells on retinal flatmounts. We observed a considerable decrease in Brn3a positive cells in saline-treated Wfs1 KO animals compared to saline-treated WT control rats (*p* < 0.05) (Figure 5l). The administration of liraglutide for 16 months rescued ganglion cell density in the Wfs1 KO rats (*p* < 0.01) (Figure 5l) and had no effect on WT retinas (Figure 5l). Moreover, in saline-treated Wfs1 KO animals, a positive correlation between insulin dose and retinal ganglion cells density was observed (R2 = 0.676, *p* < 0.05, Appendix A), but not in liraglutide-treated Wfs1 KO animals (R2 = 0.203, *p* > 0.05, Appendix A). Thus, the protection from retinal ganglion cell loss (Figure 5l) results from the therapeutic effect of liraglutide and not from the supportive treatment of insulin.

Altogether, TEM and MRI studies of optic nerve integrity and histological assessment of retinal ganglion cell survival indicated that early intervention and long-term treatment liraglutide treatment had a neuroprotective effect on maintaining the Wfs1 KO rat’s optic nerve integrity.

## 4. Discussion

Wolfram syndrome (WS) is a spectrum disorder that is affecting almost every organ system. Therefore, it is necessary to choose treatment options that are as broad as possible [39]. GLP1 RA might provide a good candidate as the receptors for GLP1 have been found throughout the organism, overlapping with tissues expressing WFS1 [40,41]. Previously we found that a 19-weeks preventive liraglutide treatment protected Wfs1 KO rats against the development of glucose intolerance [21]. Furthermore, late intervention with liraglutide protected against the progression of optic nerve atrophy, retinal ganglion cell death and protected against the decline of visual acuity [19,20]. However, it is still unknown whether liraglutide has a preventive or only delaying effect on previously mentioned symptoms in WS rats.

Therefore, the aim of this study was to evaluate the efficacy of long-term GLP1 RA treatment in a rat model of WS. For this purpose, 2-month-old WS rats without any major WS symptoms were treated with liraglutide (0.4 mg/kg/day) up to the age of 18 months and changes in diabetes markers, visual acuity and hearing sensitivity were monitored in vivo during the course of the 16-month-treatment period.

Firstly, we evaluated diabetes markers throughout the experiment. In the beginning of the treatment, the blood glucose, insulin, c-peptide and glucagon levels were indifferent between genotype and treatment groups. By the age of 7.5 months, all of the saline-treated Wfs1 KO rats developed glucose intolerance. Progressive decline in beta-cell function in Wfs1 KO animals eventually led to insulin deficiency and hyperglycemia at the age of 11.5 months: a key feature of type 2 diabetes. Whereas, Wfs1 KO animals receiving liraglutide could be divided into two distinct groups at 7.5 months of age with 63% of the cases responding to the treatment preventing the development of glycose intolerance. These two groups (glucose tolerant and intolerant, or responders and nonresponders to liraglutide) of Wfs1 KO animals in the liraglutide treatment group, could be the result of developing a tolerance to the GLP-1 receptor agonist during repetitive administration. This phenomenon has been observed in rodents, yet might not occur in humans given that chronic liraglutide administration has been shown to be as effective as acute administration [26,40,42]. It should be emphasized that during the experiment the administered dose of liraglutide remained constant: whether increasing the therapeutic dose to nonresponding animals could have improved glucose metabolism remains obscure. Nevertheless, liraglutide-treated Wfs1 KO animals were able to secrete insulin and c-peptide and we observed no indication of hyperglucagonemia. Until this time point, the glycemic control and development of diabetes followed the results of our previous study [21]. These findings suggest a sensitive timeframe in order to begin the treatment of diabetic phenotype as later intervention had minimal effect on diabetic markers in animals that had already developed glucose intolerance [19].

At the age of 11.5 months, we observed increased fasting blood glucose levels in saline-treated Wfs1 KO animals. In order to manage and subsequently minimize complications induced by hyperglycemia (i.e., diabetic retinopathy), the animals in need received insulin from 11.5 months of age [43]. By the age of 16 months, 90% of saline-treated Wfs1 KO animals had developed hyperglycemia and needed supportive insulin administration. The prevalence of diabetes in WS rats at this age is rather parallel with observations in humans diagnosed with WS [39]. Development of hyperglycemia in liraglutide-treated rats was evidently delayed. Notably, at the end of the experiment, only 55% of the Wfs1 KO animals in the liraglutide treatment group required supportive insulin treatment. These findings are supported by the clinical data where GLP-1R agonists have been shown to improve and preserve beta-cell function in patients with type 2 diabetes [44,45]. Thus, we can conclude that liraglutide treatment was able to prevent the onset of diabetes in 55% Wfs1 KO animals and delay the onset of diabetes in 45% Wfs1 KO animals.

Hearing loss has not been previously studied in a WS animal model, therefore our aim was to determine whether WS rats develop hearing loss similarly to WS patients’ and whether liraglutide treatment has an effect on it. For that purpose, we used a noninvasive DPOAE analysis, which has been effective in identifying patients with sensorineural hearing loss [46]. The DPOAE method was used to evaluate cochlear function throughout the experiment. At the age of 2 months, hearing sensitivity was indistinguishable between experimental groups. From 6.5 months, hearing sensitivity in the range from 2 to 4 kHz started to rapidly decline in saline-treated Wfs1 KO animals and this decline is further exacerbated at the age of 11 months and 17 months. Although we did not observe a positive effect of liraglutide treatment on hearing loss, our findings provide first evidence that Wfs1 KO animals develop progressive low-frequency neurosensory hearing impairment. Furthermore, we observed the decline in the hearing sensitivity at frequencies that recapitulate observations in humans [47,48,49,50]. As the frequency range of rat hearing (250 Hz to 80 kHz) has the greatest sensitivity occurring between 8 and 38 kHz, which is much higher than that found in humans (20 Hz to 20 kHz) [51], it is unfortunate that the DPOAE method was unable to measure the frequency range between 8 and 38 kHz. Nevertheless, the auditory system of rats and humans shares several anatomical and physiological features [51] and additionally, the cochlea structures of rodents and primates display a strong expression of WFS1 in homologous compartments [12,52]. Thus, our results indicate that rats develop progressive, low-frequency neurosensory hearing loss similarly to WS patients, and also position the Wfs1 KO rat as a genetic animal model for studying progressive sensorineural hearing impairment.

Ophthalmologic manifestations occur in ~90% of WS patients with median age of 11 years [39,53]. These include optic nerve atrophy, cataract, abnormal pupillary light reflexes, nystagmus, and deficiencies in visual field leading to a progressive decrease in visual acuity and impairment in color vision [1,54,55,56]. Additionally, we have previously revealed increased retinal ganglion cell death, decreased visual acuity and clear degenerative processes in Wfs1 KO rats’ optic nerves and resultant atrophy [18,19,20]. Therefore, Wfs1 KO rats can be used to evaluate the effect of liraglutide treatment on different levels in the visual pathway. First, in order to assess visual function, i.e., visual acuity, we performed the optomotor response test and observed a progressive decrease in visual acuity in the saline-treated Wfs1 KO rats. Treatment with liraglutide resulted in optomotor response values comparable to saline-treated WT rats throughout the observed timeframe. Notably, as the optomotor response test is a behavioral measurement, the deficits in visual acuity of the saline-treated Wfs1 KO could be caused by possible deficiencies in animal behavioral parameters or general well-being [57]. We cannot exclude the decline in visual acuity in saline-treated Wfs1 KO rats as being due to severe progression of cataracts. Interestingly, insulin treatment of Wfs1 KO animals had no inhibitory effect on cataract progression as described in streptozotocin-diabetic rats, suggesting a different etiology [58]. The development of cataracts seems to be independent from IOP as we did not find significant changes or abnormalities in the IOP in any treatment groups. Nevertheless, liraglutide treatment of Wfs1 KO rats’ delayed cataract progression and reduced the severity of lens opacification and was not dependent on insulin dose.

Visual system integrity was further assessed by measuring the density of retinal ganglion cells. By counting Brn3a positive cells, we confirmed a severe decrease in retinal ganglion cells in saline-treated Wfs1 KO rat retinas similar to the results obtained by optomotor response test. The 16-month treatment with liraglutide rescued the ganglion cell density in the Wfs1 KO rats and was not dependent on insulin dose. These results are in accordance with our previous observations, which showed that late intervention with liraglutide rescued ganglion cell density in Wfs1 KO rats [19]. In summary, in the current study, we observed that the treatment with liraglutide was able to halt visual pathway neurodegeneration. These results are in accordance with previous studies where GLP1 RA displayed a neuroprotective effect in Wfs1-deficient rats and other models of neurodegenerative diseases [19,20,59,60,61,62]. Hence, preventive treatment before the onset of ocular pathology with liraglutide is crucial, not only to avoid the progression of optic nerve atrophy but also to maintain retinal ganglion cell density and avoid severe cataracts in WS rat model.

## 5. Conclusions

We investigated the long-term effect of liraglutide treatment on diabetes, visual system function and integrity, and hearing sensitivity. The administration of the GLP-1R agonist liraglutide before the onset of WS symptoms offered great protection against WS progression. Results from current and previous studies indicate that preventive and long-term treatment postpones the development of glucose intolerance (as opposed to late intervention) [18] and protects against vision loss [19,20] current paper in an animal model of WS. Our results might predict possible GLP-1R agonist treatment outcomes in clinical studies on WS patients. Administration of the GLP-1R agonist liraglutide before the onset of WS symptoms offers great protection against disease progression by postponing the development of glucose intolerance (as opposed to late intervention) [18] and protecting against vision loss [19,20] current paper in an animal model of WS. The current study might provide predictive value in order to estimate GLP-1R agonist treatment outcomes in clinical studies on WS patients. The obtained results suggest that patients carrying pathogenic variants of WFS1 should be diagnosed and treated before any clinical symptoms such as diabetes and loss of vision occur as it could delay syndrome progression and therefore improve the quality of life and increase the number of healthy life years.

## Figures and Tables

**Figure 1 cells-10-03193-f001:**
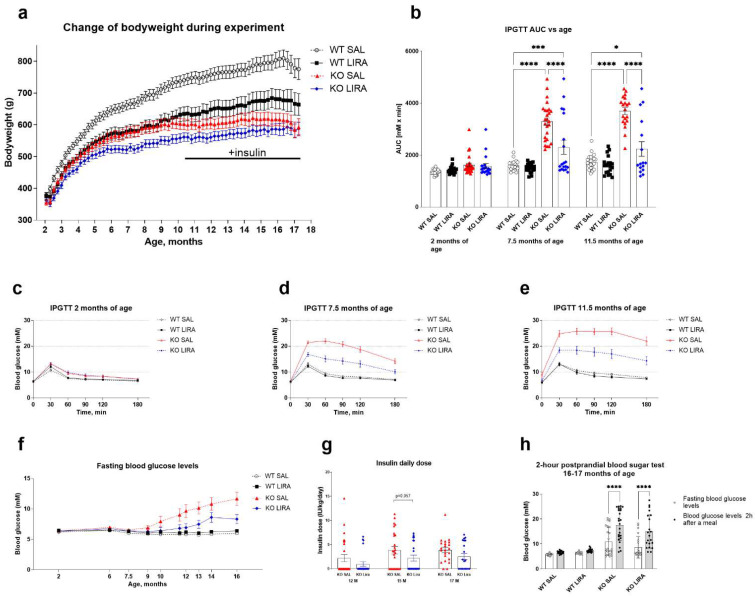
A 16-month treatment with liraglutide delays the onset of diabetes in a rat model of Wolfram syndrome. (**a**) Body weight change over 16 months of liraglutide treatment. For intraperitoneal glucose tolerance tests (IPGTTs), blood glucose levels were measured after administration of glucose (2 g/kg i.p.), and the area under the curve (AUC) was calculated. (**b**) IPGTT AUC at the age of 2, 7.5 and 11.5 months. Liraglutide treatment delays the progression of glucose intolerance in Wfs1 KO rats. (**b**) Visually, the animals in the KO LIRA group are divided into two groups, one group with glucose tolerance similar to that of WT SAL animals and the second group with animals similar to that of KO SAL. The data were compared using two-way ANOVA followed by Tukey’s multiple comparisons test. The data are presented as the mean ± SEM, * *p* < 0.05, *** *p* < 0.001, **** *p* < 0.0001, *n* = 16–27 per group. IPGTT at the age of (**c**) 2, (**d**) 7.5 and (**e**) 11.5 months. (**f**) Fasting blood glucose levels during 16-month treatment. (**g**) Insulin daily dose at 12, 15 and 17 months of age, the data were compared using nonparametric Mann−Whitney test. The data are presented as the mean ± SEM, *n* = 24–27 per group. For 2 h postprandial blood sugar tests, blood glucose levels were measured after 3 h of fasting and then blood glucose was measured 2 h after the animals had access to food. (**h**) Blood glucose levels from 2 h postprandial blood sugar test at the age of 16–17 months, the data were compared using repeated measures ANOVA followed by Bonferroni’s multiple comparisons test. The data are presented as the mean ± SEM, **** *p* < 0.0001, *n* = 21–24 per group.

**Figure 2 cells-10-03193-f002:**
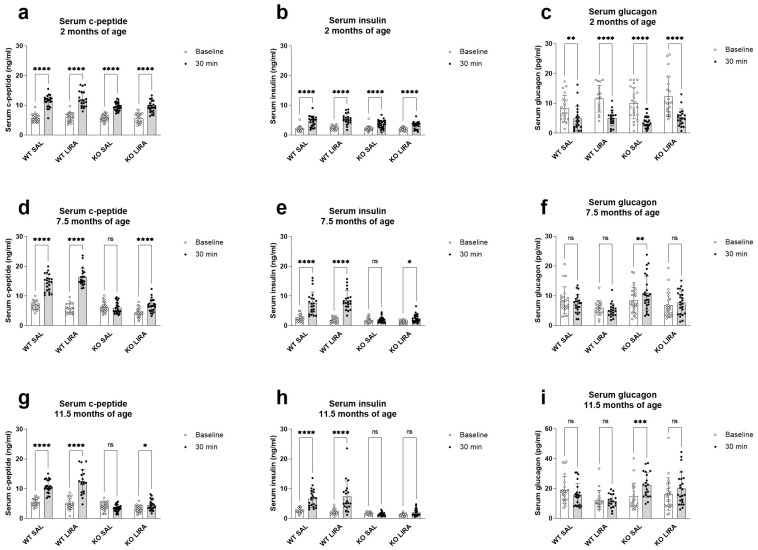
Prolonged treatment with liraglutide maintains c-peptide secretion until the age of 11 months and thereby delays the development of diabetes in a rat model of Wolfram syndrome. At the age of 2, 7.5 and 11.5 months, serum (**a**,**d**,**g**) c-peptide, (**b**,**e**,**h**) insulin and (**c**,**f**,**i**) glucagon levels before and 30 min after administration of glucose in the glucose tolerance test (IPGTT). (**a**–**c**) At the age of 2 months, there were no differences between experimental groups. At the age of 7.5 months, Wfs1 KO rats show the first signs of diabetes, i.e., reduced (**d**) c-peptide and (**e**) insulin levels, and increased (**f**) glucagon levels after glucose administration. Liraglutide treatment prevented these changes and at the age of 11.5 months, liraglutide-treated Wfs1 KO rats were still able to secrete (**g**) c-peptide. The data were compared using repeated measures ANOVA followed by Bonferroni’s multiple comparisons test. The data are presented as the mean ± SEM, non-significant (ns), * *p* < 0.05, ** *p* < 0.01, *** *p* < 0.001, **** *p* < 0.0001, *n* = 20–27 per group.

**Figure 3 cells-10-03193-f003:**
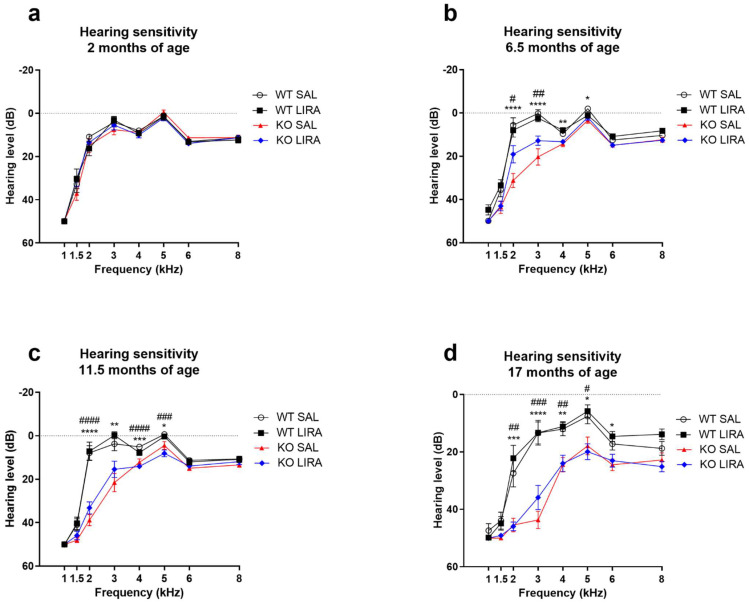
At the age of 6.5 months, hearing sensitivity of Wfs1 KO rats starts to decline in the range of 2 to 4 kHz and liraglutide treatment delays the decline of hearing sensitivity. Hearing sensitivity was measured for both ears and the data were averaged. Hearing sensitivity at (**a**) 2; (**b**) 6.5; (**c**) 11.5 and (**d**) 17 months of age. (**a**) At the age of 2 months, hearing sensitivity is similar for all animals regardless of treatment group and genotype. (**b**) At the age of 6.5 months hearing sensitivity of Wfs1 KO rats starts to decline in the range of 2 to 4 kHz, this decline is further exacerbated at the age of (**c**) 11.5 months and (**d**) 17 months. (**b**) Liraglutide delays the decline of hearing sensitivity. However, liraglutide treatment does not prevent the decline of hearing sensitivity in older Wfs1 KO rats (**c**,**d**). The data were compared using one-way ANOVA followed by Tukey’s HSD tests. The data are presented as the mean ± SEM. Significance was measured between genotype and treatment. * *p* < 0.05, ** *p* < 0.01, *** *p* < 0.001, **** *p* < 0.0001 WT SAL compared to KO SAL, # *p* < 0.05, ## *p* < 0.01, ### *p* < 0.001, #### *p* < 0.0001 WT SAL compared to KO LIRA, *n* = 15−26.

**Figure 4 cells-10-03193-f004:**
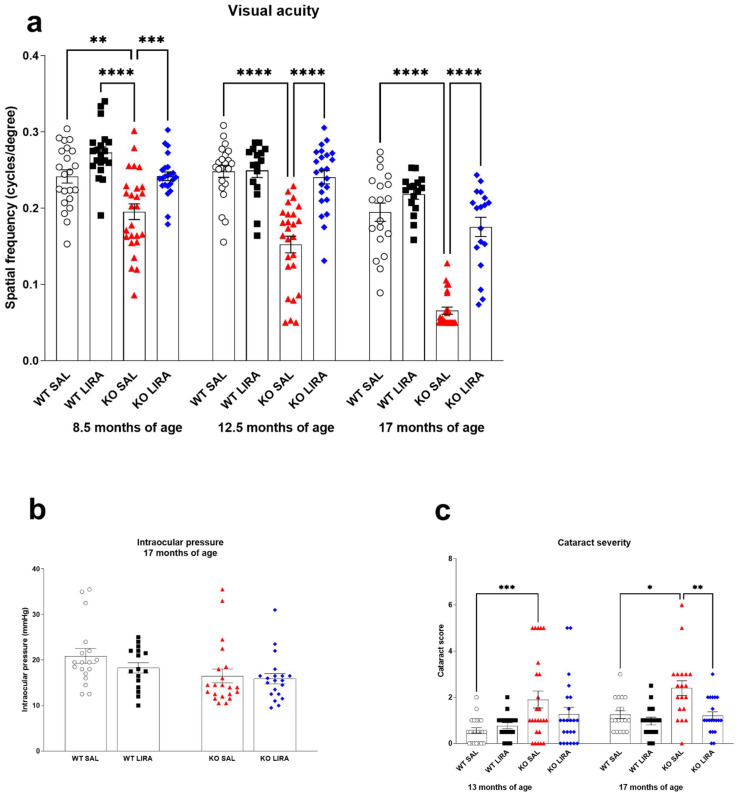
A 16-month treatment with GLP1 receptor agonist liraglutide protects against vision loss in a rat model of Wolfram syndrome. (**a**) Optokinetic reflex response at the age of 8.5, 12.5 and 17 months. Response was measured for both eyes and the data were averaged. In KO SAL animals, visual acuity decreased over time, while in KO LIRA animals, visual acuity stayed at the same level as WT SAL animals. (**b**) Intraocular pressure at the age of 17 months was unchanged between experimental groups, the data were compared using one-way ANOVA followed by Tukey’s multiple comparisons test. (**c**) Cataract severity at the age of 13 and 17 months. The data were compared using two-way ANOVA followed by Tukey’s multiple comparisons test. The data are presented as the mean ± SEM, * *p* < 0.05, ** *p* < 0.01, *** *p* < 0.001, **** *p* < 0.0001, *n* = 16–24 per group.

**Figure 5 cells-10-03193-f005:**
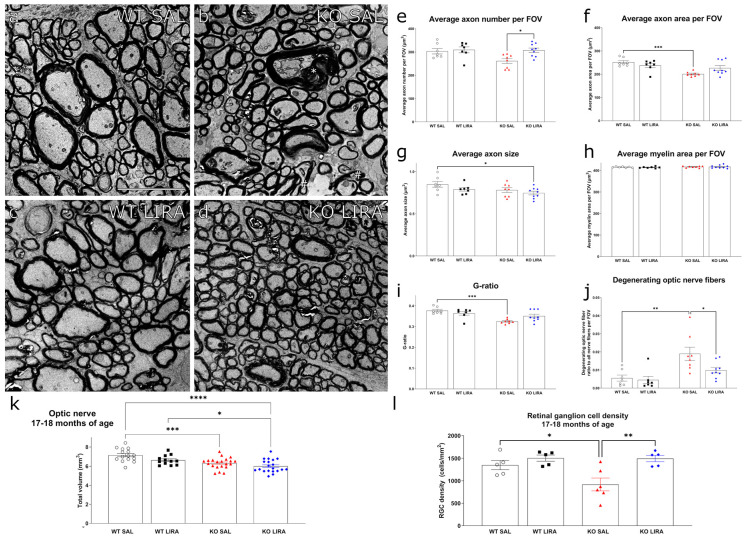
Morphometric analyses of optic nerve and evaluation of ganglion cell density. (**a**–**d**) Electron micrographs representing optic nerves from corresponding treatment groups. (**b**) Large caliber optic nerve fibers (asterisk) display several degeneration signs: severe vacuolization and hypermyelination. Axonal compression (hash) is observable throughout the optic nerve. (**e**–**l**) Graphs displaying measurements of: (**e**) average number of axons per field of view (FOV); (**f**) average area of axons (μm^2^) per field of view (FOV); (**g**) average axon cross-section area (**h**) average myelin area (μm^2^) per FOV; (**i**) area based G-ratio (axon area divided by axon area plus myelin area); (**j**) ratios of degenerating fibers calculated as the ratio of degenerating fibers to all counted fibers per FOV, *n* = 7–9 per group. (**k**) Volume of optic nerve obtained by analyzing MRI data at 17–18 months of age, *n* = 12–21 per group. (**l**) Graph displays retinal ganglion cell density at the end of the experiment, at the age of 17–18 months, *n* = 5–6 per group. Quantitative data were compared one-way ANOVA followed by Tukey’s multiple comparisons test. The data are presented as the mean ± SEM, * *p* < 0.05, ** *p* < 0.01, *** *p* < 0.001, **** *p* < 0.0001. Scale bar: 5 µm.

## Data Availability

Please contact corresponding authors for data requests.

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
