# Peer review of "Early Intervention and Lifelong Treatment with GLP1 Receptor Agonist Liraglutide in a Wolfram Syndrome Rat Model with an Emphasis on Visual Neurodegeneration, Sensorineural Hearing Loss and Diabetic Phenotype"

_cells, 2021, doi:10.3390/cells10113193_

Round 1

Reviewer 1 Report

Jagomäe et al. reported the protective effects of 16-month-long treatment with GLP1 receptor agonist liraglutide in a Wolfram Syndrome (WS) rat model. They found that the treatment with liraglutide delayed the onset of diabetes and protected against vision loss in a rat model of WS, although it did not prevent progressive sensorineural hearing loss in this model. The authors argued that early diagnosis and prophylactic treatment with the liraglutide may also prove to be a promising treatment option for WS patients by increasing the quality of life of WS patients. This is a continuous study from this group to further characterize this rat model of WS and to investigate the protective effects of long-term treatment of liraglutide in this model. Overall, this study is interesting and may provide potential insights for using liraglutide to treat the WS patients. My comments are listed below.

  1. Liraglutide has been found to have multiple side effects in clinic. Did the authors notice any side effects of this life-long treatment in the rat model of WS? Appropriate considerations should be discussed at least. Also, injections of 1 ml/kg volume were made subcutaneously once a day for 16 months. What’s the rationale of using this dosage and frequency for liraglutide? Is it possible to reduce the frequency of injections or increase the dosage with much less frequency to reduce the potential side effects and stress to the animals?
  2. It seems that there is a gender difference in terms of the severity of phenotypes in WS mice. Have the authors seen this difference in their rat model, too? It’s better to include this in their discussion. If only used males, it should be justified, too.
  3. In some aged Wfs1 knockout mice, their eyes popped out or ruptured at late stages. Did the authors see similar phenomena in aged Wfs1 KO rats?
  4. In Figure 5a, the authors should show representative images of 4 groups and using arrows to indicate corresponding
  5. In Figure 5i, the Brn3a staining in retinal ganglion cells in 4 groups should be
  6. In Methods, is there any particular reason of not transferring to 30% sucrose after 24-h fixation of optic nerve? Instead, the authors let the optic nerve sit in 1x PBS for three more days.
  7. The first few sentences in the section of Discussion are
  8. The last two sentences in the section of Discussion should be
  9. There are a few typos need to be corrected, such as animas, andlate, etc.

Author Response

Point-by-point response in attached file

Reviewer 2 Report

The work presented discusses a relevant topic, namely the Wolfram Syndrome, and extends our knowledge on liraglutide as a potential treatment for the disease. This study is clearly the result of a long, extensive work with much data supporting the conclusions.

However, I consider the MS to have some major and minor flaws that I will list below:

  • The authors spread some key facts about the Wolfram Syndrome in the Results and Discussion sections that make it difficult to understand the progression of the disease in humans, and by consequence to assess the relevance of the animal model and timepoints chosen. Some of them are not consistent. Moving those facts together to the Introduction would help the reader greatly. Examples below:
    1. in l.38:” The syndrome first manifests as diabetes mellitus, followed by optic nerve atrophy, deafness and neurodegeneration”
    2. 443:” The first symptom of WS is considered insulin dependent diabetes”
    3. 425:” Diagnosis of WS is usually made during adolescence when aforementioned core clinical signs of the syndrome become evident”
    4. And in l. 481:” Sensorineural hearing loss in WS patients is slowly progressing and is typically diagnosed at the age of 8 years, with 75% prevalence [42].”
  • I found it difficult to understand the main objective of the study in the introduction because of two facts:
    1. Line 67. “the aim of this study was to evaluate the safety and efficacy of long- term,”
    2. Paragraph from line 58 to line 62 mentions that previous studies have already tested liraglutide in this very same animal model.

First, there is no safety evaluation in this work and that should be removed from line 67.

Secondly, it is only in Line 338-341 that the authors clearly refer one conclusion of a previous study from the same group. This piece of information and the conclusions of the group’s previous works should be consistently explained and connected in the introduction so the objective becomes clear. As is, I struggled to find the novelty of the work and what it is building upon.

  • Part of the experimental design raises some questions.
    1. Fact 1: SOME animals received supportive insulin treatment (l.99)
      1. It is important to discuss the effect this insulin treatment may have on each of the results. For example, Can this insulin have protected the animals from developing earing loss? Or is it possible that insulin treatment may aggravate the progression of the disease? (you compare this model to Type 2 diabetes in l.449)
    2. Fact 2: (l.107) many animals died during the experiment (up to 40%! per group), mostly of cancer. This fact is not mentioned again in the MS but I think it should be discussed. Were KO or LIRA groups affected any differently from controls? Were the surviving animals healthy or did most of them also have cancer (and just didn’t die yet)?
    3. Finally, putting these two facts together, wouldn’t it have been more appropriate to end the experiments before the animals were 18 months of age? Would that have avoided this high dead rate and the insulin treatment which can be two important confounding factors? (performing a survival curve analysis for both insulin injection and death might help answer most of these questions)
  • GLP1 RA is a key player in this study through the action of liraglutide, however its relation to the syndrome is barely discussed. I consider it would be important to add a section explaining the link of GLP1 to the symptoms of Wolfram Syndrome, and discuss its relation to Wfs1.

Minor:

  • L58: this is the first time GLP1 is mentioned in the main text (besides abstract) and should be spelled
  • L70-72: “We found…vision loss.”; This sentence should be removed as it is a conclusion, not part of the background leading to the present work.
  • L91: “WT Lira”. “Lira” is spelled with only first capital letter, while with all capital letters in other parts of the text. Consistency should be kept throughout.
  • L256: in 63% of Wfs1 KO animals (p < 0.0001) (Fig-256 ure 1b, e). By 11.5 months of age, 62.5% of liraglutide-treated Wfs1 KO animals remained 257 diabetes-free (p<0.0001) (Figure 1b) whereas”
    • Although these frequencies are mentioned, no statistical test seems to have been performed. By the statistical description in image 1 and in the Methods section, only the means were compared.
  • 1g: I would seriously consider performing a non-parametric analysis as distributions are clearly not gaussian.
  • 372: “The number of optic nerve axons area per FOV remained”. Remove “number of”.
  • 5a: we do not know which image corresponds to each experimental group as only KO SAL is identified.
  • 420: why perform ONE-way ANOVA when a two-way ANOVA would be able to analyze the two independent factors of the data of all graphs in figure ?
  • 440: 0,4 to 0.4
  • 557:” Authors should discuss the results and how they can be interpreted from the perspective of previous studies and of the working hypotheses. The findings and their implications should be discussed in the broadest context possible. Future research directions may also be highlighted.” Please, remove.

Author Response

(The authors gave the same response as above.)

Round 2

Reviewer 1 Report

The authors improved the manuscript based on the reviewers' comments. However, I did not find the representative images of Brn3a staining in retinal ganglion cells in 4 groups, although the authors claimed the manuscript was revised accordingly in the response letter. Also, the typo of animas in line 167 has not been corrected.

Author Response

Dear reviewer 1,

We thank you for your valuable comments and time.

As you suggested, we added the representative images of Brn3a staining in retinal ganglion cells in 4 groups to the supplementary file (Supplementary figure 3). Additionally, the typo in line 167 has been corrected.

Moreover, we added to the supplementary file the correlation between insulin dose (IU/kg/day) and cataract score (Supplementary Figure 1) and the correlation between insulin dose (IU/kg/day) and RGC density (cells/mm 2) (Supplementary Figure 2). These figures confirm that the protection from cataract (Figure 4c) and from retinal ganglion cell loss (Figure 5l) result from the therapeutic effect of liraglutide and not from the supportive treatment of insulin.

All changes made in the manuscript are highlighted with blue. The changes do not affect the conclusions of the study in any way.

Reviewer 2 Report

I would like to thank the authors for clarifying some of the doubts I had about the MS.

I also consider the issues to be tackled.

Author Response

Dear reviewer 2,

We thank you for your valuable comments and time.

We added the supplementary file, where we show the correlation between insulin dose (IU/kg/day) and cataract score (Supplementary Figure 1) and the correlation between insulin dose (IU/kg/day) and RGC density (cells/mm 2) (Supplementary Figure 2). These figures confirm that the protection from cataract (Figure 4c) and from retinal ganglion cell loss (Figure 5l) result from the therapeutic effect of liraglutide and not from the supportive treatment of insulin.

Moreover, we added the representative images of Brn3a staining in retinal ganglion cells in 4 groups to the supplementary file (Supplementary figure 3).

All changes made in the manuscript are highlighted with blue. The changes do not affect the conclusions of the study in any way.